# Effect of community-based newborn care implementation strategies on access to and effective coverage of possible serious bacterial infection (PSBI) treatment for sick young infants during COVID-19 pandemic

**Gizachew Tadele Tiruneh**[1]*, **Nebreed Fesseha**[1], **Dessalew Emaway**[1],
**Wuleta Betemariam**[2], **Tsinuel Girma Nigatu**[3], **Hema Magge**[4,5], **Lisa Ruth Hirschhorn**[6]

1 JSI Research & Training Institute Inc., Addis Ababa, Ethiopia, 2 JSI Research & Training Institute Inc., Washington, DC, United States of America, 3 Department of Pediatrics and Child Health, Jimma University, Jimma, Ethiopia, 4 Bill & Melinda Gates Foundation, Seattle, Washington, United States of America, 5 Ethiopia and Fenot Project—School of Population and Public Health, University of British Columbia, Vancouver, BC, Canada, 6 Feinberg School of Medicine and Havey Institute of Global Health, Northwestern University, Chicago, Illinois, United States of America

* gizt121@gmail.com

## Abstract

### Background

In Ethiopia, neonatal mortality is persistently high. The country has been implementing community-based treatment of possible serious bacterial infection (PSBI) in young infants when referral to a hospital is not feasible since 2012. However, access to and quality of PSBI services remained low and were worsened by COVID-19. From November 2020 to June 2022, we conducted implementation research to mitigate the impact of COVID-19 and improve PSBI management implementation uptake and delivery in two woredas in Ethiopia.

### Methods

In April-May 2021, guided by implementation research frameworks, we conducted formative research to understand the PSBI management implementation challenges, including those due to the COVID-19 pandemic. Through a participatory process engaging stakeholders, we designed adaptive implementation strategies to bridge identified gaps using mechanism mapping to achieve implementation outcomes. Strategies included training and coaching, supportive supervision and mentorship, technical support units, improved supply of essential commodities, and community awareness creation about PSBI and COVID-19. We conducted cross-sectional household surveys in the two woredas before (April 2021) and after the implementation of strategies (June 2022) to measure changes in targeted outcomes.

**Data Availability Statement:** The dataset used and analyzed during this study is included as supplementary information to this article (Additional file 3).

**Funding:** The article write-up and publication fee were supported by the Bill & Melinda Gates Foundation, Investment Number INV-024241. JSI Research & Training Institute, Inc. has helped us in the form of salaries for authors [GT, NF, DE, and WB]. However, none of the funders played a role in the study design, data collection, analysis, decision to publish, or manuscript preparation.

**Competing interests:** The authors declare that they have no competing interests. The authors [GT, NF, DE, and WB] work for JSI Research & Training Institute, Inc., a commercial company. This does not alter our adherence to PLOS ONE policies on sharing data and materials. One of the authors of this manuscript (HM) works for the Gates Foundation. We would like to declare that we do not have any conflict of interest with the Gates Foundation-paid staff in preparing this manuscript.

**Abbreviations:** BMGF, Bill & Melinda Gates Foundation; CFIR, Consolidated Framework for Implementation Research; CHIS, Community Health Information System; COVID-19, Coronavirus; EPOC, Effective Practice and Organization of Care; ERIC, Expert Recommendations for Implementation Change; HEP, Health Extension Program; HEW, Health Extension Worker; iCCM, integrated community case management of childhood illnesses and newborn care; IDI, in-depth interview; IRLM, Implementation Research Logic Model; JSI, JSI Research & Training Institute Inc.; KOICA, Korea International Cooperation Agency; L10K, Last Ten Kilometers Project; LMIC, low-and-middle-income countries; NMR, neonatal mortality rate; PSBI, possible serious bacterial infection; RE-AIM, Reach, Effectiveness, Adoption, Implementation, and Maintenance; SDG, sustainable development goal; USAID, United States Agency for International Development; WDA, Women Development Army; WHO, World Health Organization.

## Results

We interviewed 4,262 and 4,082 women who gave live birth 2–14 months before data collection and identified 374 and 264 PSBI cases in April 2021 and June 2022, respectively. The prevalence of PSBI significantly decreased (p-value = 0.018) from 8.7% in April 2021 to 6.4% while the mothers' care-seeking behavior from medical care for their sick newborns increased significantly from 56% to 91% (p-value <0.01). Effective coverage of severely ill young infants that took appropriate antibiotics significantly improved from 33% [95% CI: 25.5–40.7] to 62% [95% CI: 51.0–71.6]. Despite improvements in the uptake of PSBI treatment, persisting challenges at the facility and systems levels impeded optimal PSBI service delivery and uptake, including perceived low quality of service, lack of community trust, and shortage of supplies.

## Conclusion

The participatory design and implementation of adaptive COVID-19 strategies effectively improved the uptake and delivery of PSBI treatment. Support systems were critical for front-line health workers to deliver PSBI services and create a resilient community health system to provide quality PSBI care during the pandemic. Additional strategies are needed to address persistent gaps, including improvement in client-provider interactions, supply of essential drugs, and increased social mobilization strategies targeting families and communities to further increase uptake.

## Introduction

Sub-Saharan Africa had the world's highest neonatal mortality rate (NMR) at 27 deaths per 1,000 live births in 2021 [1]. In Ethiopia, the NMR is a major public health problem that has not significantly declined since 2000, and currently stands at 33 per 1,000 live births, contributing to about 56% of under-five child mortality [2]. Considering the slow deceleration rate of NMR and the impact of COVID-19, most sub-Saharan African countries, including Ethiopia, are at risk of missing the Sustainable Development Goal (SDG) target for neonatal mortality [1].

Severe neonatal infections are among the leading causes of death accounting for half a million newborn deaths in 2019 [3]. In resource-limited settings, access to inpatient treatment is difficult for many young infants with signs of severe infection due to limited accessibility to acceptable or affordable care [4, 5]. Informed by evidence from low-and-middle-income countries (LMICs) that demonstrated the effectiveness of simplified regimens, including injectable and oral antibiotics, the World Health Organization (WHO) recommends the management of possible serious bacterial infection (PSBI) outside of the hospital setting when referral is not feasible [6].

Ethiopia adopted community-based newborn infection management in 2012 and incorporated it within the integrated community-based management (iCCM) of common childhood illnesses platform to improve the use of iCCM services for newborns and young infants by Health Extension Workers (HEWs) when referral is not feasible [7]. However, the crude and effective coverage of PSBI management remains low. The COVID-19 pandemic [8, 9] and multiple pre-existing contextual factors resulted in increased barriers to uptake and delivery. These included erratic supply of essential drugs and poor logistics management systems [10,

11]; suboptimal supportive and referral links [10–12]; suboptimal quality of care [13]; community misperceptions about newborn illnesses [14, 15]; inaccessibility of facilities; and socio-cultural beliefs [10, 15] which all affected the implementation and uptake of iCCM services including PSBI diagnosis and treatment [16].

Improving both access to and quality of PSBI management is needed to address barriers to effective improvement of newborn health outcomes. However, expanding access and delivery without focusing on quality is not sufficient to achieve the intended outcomes. Evidence in LMICs showed that effective quality coverage is considerably lower than coverage for maternal and child health services ignoring quality [17]. As such, the global health community recognized high-quality health care as an important component of efforts to reach SDG 3 for the achievement of universal health coverage [18, 19]. Monitoring trends in effective coverage of PSBI management and place of treatment (health facility or community) is crucial to assess the quality of community-based iCCM programs. However, in Ethiopia, population-level data on the effective coverage of PSBI treatment and place of treatment are scarce.

Between November 2020 and June 2022, the Bill & Melinda Gates Foundation (BMGF) funded the JSI Research & Training Institute Inc. (JSI) Last Ten Kilometers (L10K) Project to conduct implementation research (IR) in Ethiopia to design and support strengthening the adaptive implementation of community-based management of PSBI for sick young infants and to mitigate the impact of COVID-19 on neonatal mortality. Employing an implementation science evaluation framework, we evaluated the effects of these health systems and community strategies on community-based neonatal sepsis management on access to and effective coverage of PSBI for sick young infants.

## Methods

### Study context and settings

Ethiopia's primary-level care system includes a primary hospital, health centers, and health posts; supported by secondary and tertiary-level care. The primary health system in a typical woreda (i.e., district) should have one primary hospital, 4–5 health centers, 20–25 health posts, 40–50 Health Extension Workers (HEWs), and about 4,000 Women Development Army (WDA) volunteers to serve a population of approximately 100,000 people [20]. Primary health care services include promotive, preventative, curative, and rehabilitative services at facilities (i.e., at health posts, health centers, and primary hospitals), at home or in the community, outreach services through the Health Extension Program (HEP), and the engagement of community volunteers. Health posts are staffed with two female HEWs and supported by a network of WDAs providing promotive maternal and newborn services and some basic antepartum and postpartum counseling and referral services.

Health Extension Workers and WDAs are responsible for educating mothers about early recognition of illness in newborns, and prompt care-seeking assessing danger signs in newborns during postnatal home visits [21], In addition, community-based treatment of PSBI provided by HEWs, integrated with the iCCM program. Health Extension Workers are trained to identify newborn infections and provide treatment when referral is not feasible.

This implementation research was conducted at Dembecha and Lume woredas of the Amhara and Oromia regions. The two woredas are comprised of 11 health centers and 66 health posts that serve an estimated 250,000 rural people.

## Participatory design of strategies and pathways of adaptive implementation

We employed IR frameworks in designing, implementing, and evaluating the adaptive implementation of PSBI treatment during COVID-19 in Ethiopia. The Reach, Effectiveness, Adoption, Implementation, and Maintenance (RE-AIM) [22] and Consolidated Framework for Implementation Research (CFIR) [23] frameworks were used to define outcomes and identify factors that needed to be addressed. Fig 1 below depicts the simplified process of the implementation research to explore the different dimensions of RE-AIM.

In April 2021, we conducted a narrative review of existing literature followed by formative research to identify implementation challenges. Details of the findings are presented elsewhere [16]. In May 2021, we facilitated a participatory design of strategies by engaging relevant stakeholders to systematically develop, identify, categorize, and prioritize implementation strategies to address identified and newly emerged barriers. Then, we matched our implementation strategies with the Effective Practice and Organization of Care (EPOC) [24] and Expert Recommendations for Implementing Change (ERIC) [25] taxonomies and strategies identified from previous implementation research on the management of PSBI when referral was not feasible at the individual, community, facility, and system levels [4, 12, 26–33]. At the individual level, we identified on-site coaching and training of HEWs by PSBI/iCCM-trained focal mentors from health centers as strategies. At the facility and systems level, we identified the following strategies: coordinated referrals between health posts and health centers; the establishment of a technical support unit to facilitate implementation across the area; strengthening of the supply chain, support system, and links; integration of management of PSBI implementation in the woreda health system work stream; and integration of COVID-19 and routine services (Table 1). Details of the implementation challenges, strategies, mechanisms of action, and their outcomes can be found elsewhere [16].

The strategies were implemented effectively, ensuring the continued delivery of PSBI services while improving mothers' care-seeking behavior for their sick young infants. Technical

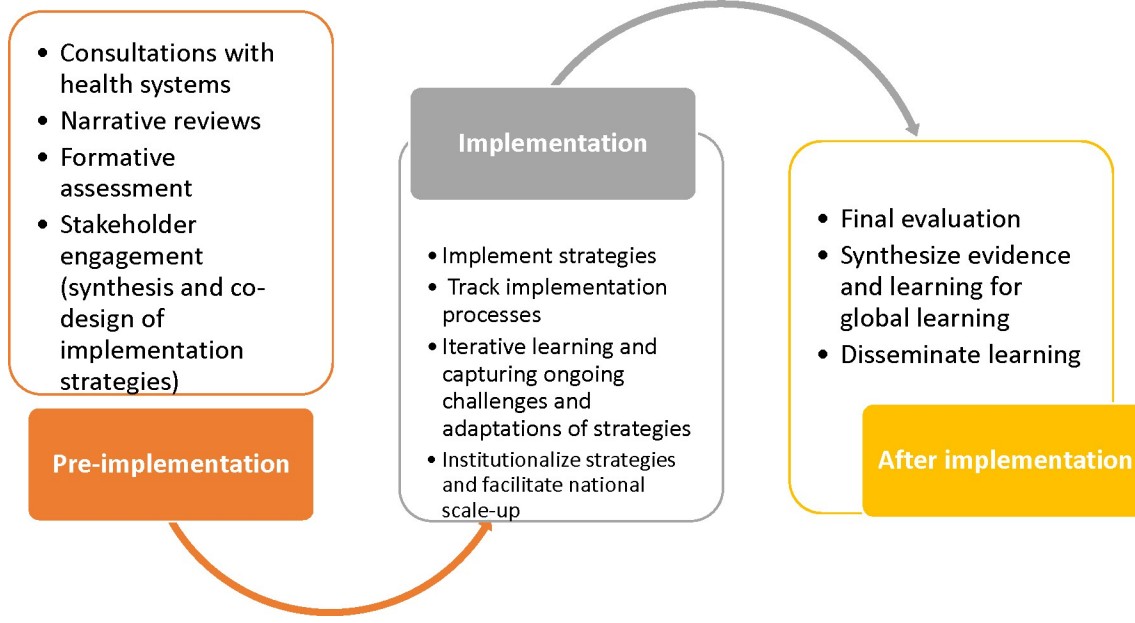

**Fig 1. Program design and implementation process, 2021.**

**Table 1. Implementation strategies for PSBI treatment when a referral is not possible.**

| Challenges | Strategies | Details of activities and roles of stakeholders |
|---|---|---|
| Low confidence and competency of HEWs | On-site coaching and training of HEWs | iCCM-trained focal mentors from health centers (HC) provide quarterly coaching to HEWs on PSBI treatment including gentamicin injection |
| | Organize offsite training for HEWs and their supervisors not trained on basic PSBI treatment | JSI facilitated seven days of basic iCCM training to HEWs and their supervisors who were not previously trained |
| | Facilitate PHCU-level performance review | The health center director facilitated monthly PHCU-level performance review meetings amongst all HEWs and HC staff |
| Weak health system support | Establish a technical support unit for iCCM technical oversight (assign a focal person and trained supervisors) | The project facilitated the establishment of technical support units involving PHCU director and WorHO child health officers to provide technical oversight to the iCCM program and HEWs |
| | Strengthen support systems | • iCCM trained personnel from the health centers and woreda health office conducted supportive supervision of the health posts;<br>• The project also conducted three rounds of monitoring visits to health centers and health posts in both intervention woredas |
| | Strengthen supply chain | • JSI advocated the national procurement of gentamicin 20 mg/ml;<br>• WorHO and HC directors facilitated the redistribution of iCCM supplies;<br>• Supervisors from the project as well as from the HC support HEWs on support appropriate use of purchase request forms and bin cards |
| | Conduct advocacy meetings with woreda administrative and kebele managers | WorHO child health officers, supervisors, and HEWs conducted advocacy meetings with woreda administrative and kebele managers |
| | Strengthening referral linkage between health posts and health centers | Health center director and supervisors assign iCCM-trained staff |
| Suboptimal integration of PSBI treatment | Integrate management of PSBI in the woreda health system work stream | The project facilitated the integration of PSBI treatment into the woredas and PHC work stream (annual planning and budgeting, supervision, and performance review) |
| Sub-optimal community engagement | Awareness creation regarding available child health services at the HP level and COVID-19 | • HEWs community dialogues with kebele leaders, WDAs, and kebele-level multi-sectoral meetings to create demand for PSBI management;<br>• HEWs, community volunteers, and kebele managers conducted community awareness creation activities using different community gatherings;<br>• HEWs and HC staff conducted continuous community education about the method of transmission and prevention mechanisms of COVID-19 to communities |

HC: health center; HP: health post; HEW: Health Extension Worker; OPD: outpatient department; PHCU: Primary Health Care Unit; PSBI: possible serious bacterial infection; WDA: Women Development Army; WorHO: Woreda health office

support units comprised of iCCM trained personnel from under-five clinics of health centers, health center directors, and woreda health office child health officers were established. They provided on-site training and coaching to HEWs through supportive supervision, facilitated performance reviews, coordinated referrals, provided feedback to HEWs, identified implementation challenges, developed interventions in collaboration with stakeholders, and supported community sensitization efforts [16]. Monitoring data revealed that 84% of HEWs treated at least one sick young infant in the previous six months. Between April 2021 and May 2022, HEWs conducted 32,428 home visits, identifying 3,099 newborns, approximately three infants per health post per month. During the same period, HEWs assessed 1,335 severe young infant illness cases, representing an 85% coverage rate.

Implementation strategies were largely integrated into community-level healthcare worker workstreams, woreda reports, and the MOH's annual planning and forecasting of iCCM/ PSBI commodities for 2023 and 2024.

## Evaluation design

We evaluated the effects of the implementation strategies on access to and treatment cascade of PSBI using before-after household surveys. Data were collected from 18 April-24 May 2021

for the baseline and from 06 June-09 July 2022 for the end-line. In addition, at the end-line, a programmatic qualitative study was conducted to explore the perceived impact of the strategies and barriers to implementation.

## Sample size and sampling technique

The sample size for the household survey was determined using a double population formula in a comparative before and after cross-sectional study design. This would be sufficiently powered to detect a 20-percentage point change in sick newborn care-seeking due to the intervention. The assumption to calculate the sample size considered that 56% of mothers sought medical care for ill babies in the neonatal period [34], 95% confidence level (Z$\alpha$/2 = 1.96), power of 80%, and design effect of 1.5. Accordingly, 4,262 and 4,082 mothers with infants 2–14 months were interviewed during baseline and end-line, respectively from all 66 kebeles in the intervention woredas.

After obtaining an updated list of households with under-15-month-old infants using unique household identifiers from the Community Health Information System (CHIS) from all 66 health posts, we recruited all eligible respondents. For the qualitative component of the study, the theoretical sampling technique was used to collect information until saturation in each data category was achieved.

## Data collection

We conducted before-after cross-sectional household surveys of women who gave live birth 2–14 months before data collection and resided in the two woredas. We used a structured questionnaire translated into Amharic and Oromiffa local languages using a web-based SurveyCTO mHealth platform on data collector smartphones (S1 File).

During the end line, we conducted 41 in-depth interviews (IDIs) with 19 HEWs, 11 health center directors, nine health center iCCM focal persons, and two woreda child health officers using an open-ended questionnaire-guided interview that was audio-recorded (S2 File). Interviews were conducted in the Amharic and Oromiffa languages by experienced qualitative research consultants.

## Outcome measures

In this paper, effective coverage of PSBI treatment is defined as coverage of PSBI treatment adjusted for receipt of the appropriate antibiotics (gentamicin injection and/or amoxicillin) among young infants 0–59 days with signs and symptoms of severe bacterial infection (e.g., difficulty breathing, chest in-drawing, ceasing to feed well, unusually hot or cold, less active than usual, and/or convulsions) who sought care from the appropriate provider (either from health professionals or HEWs). First, we calculated the crude service contact coverage as the percent of sick young infants with PSBI symptoms who sought care from appropriate providers. Then, we calculated intervention-adjusted coverage of sick young infants with PSBI who sought care from appropriate providers who received antibiotics treatment. Finally, we calculated the effective coverage cascade, beginning with the target population, crude service contact coverage, and intervention-adjusted coverage adjusting for the recipient of appropriate treatment, excluding those who took inappropriate treatment or unnecessary antibiotics such as tetracycline eye ointments, antimalarial drugs, oral rehydration solution, vitamin A supplements, and/ or traditional remedies [35, 36].

## Data analysis

Quantitative data analysis was conducted using Stata version 15.1. Socio-demographic data were summarized by frequency tables and summary statistics. The study was adjusted for the cluster survey design effect. We estimated statistically significant ($p < .05$) differences in the prevalence of PSBI and care-seeking behavior of mothers between the baseline and the end-line surveys using chi-square statistical tests.

The audio recordings from in-depth interviews were transcribed verbatim for the qualitative data, and the transcript texts were exported to ATLAS.ti software for thematic analysis. These findings are integrated with the quantitative component of the evaluation.

## Ethics approval and consent to participate

Ethical clearance was obtained from the Ethiopian Public Health Association (EPHA) Research Ethics Review Committee (Reference #: EPHA/OG/166/21 dated April 16, 2021) and renewed for the end-line survey (Reference #: EPHA/OG/781/22 dated April 18, 2022). Written permission to undertake the study was sought from the zonal health department and institutions. Informed verbal consent and voluntary participation were ensured. All study participants were informed about the study's purpose, benefits, and hazards, including their right to opt out and to respond to questions without consequence. If the respondent was less than 18 years old, then consent was sought from her husband or guardian. We deidentified and stored respondents' information. The authors confirm that all methods were carried out in accordance with the relevant guidelines and regulations.

## Results

### Respondent characteristics

In the household survey, 4,262 and 4,081 women with children aged 2–14 months were interviewed during April 2021 and June 2022, respectively. More than two-thirds of the respondents were in the 20-34 year age group. The majority had no formal education or literacy skills (56% in April 2021 and 53% in June 2022), and about half of them lived 30 minutes or more walking distance from health posts or health centers (47% in April 2021 and 55% in June 2022).

There were differences in the background characteristics of the respondents for maternal education, number of children, and distance to health facilities. In the end-line survey, respondents were more likely better educated, had less parity, and resided within 30 minutes of the facility (Table 2).

### Changes in prevalence of PSBI and care-seeking behavior

The prevalence of any history of illness during the neonatal period was significantly reduced from 11.6% [95% CI: 9.8–13.7] in April 2021 to 8.4% [95% CI: 6.9–10.3] in June 2022. Similarly, the prevalence of PSBI reduced from 8.4% [95% CI: 7.0–10.0] in April 2021 to 6.2% [95% CI: 4.8–7.8] in June 2022. The care-seeking behavior of mothers for their severely sick neonates had significantly improved from 56.1% [95% CI: 45.3–66.4] in April 2021 to 91.3% [95% CI: 87.9–93.8] in June 2022 (Table 3).

### Change in the place and time of care-seeking and treatment

A statistically significant proportion of mothers sought care for PSBI at health posts after the intervention: 4.2% (95% CI: 2.0–9.0%) at baseline to 26.4% (95% CI: 16.4–39.5%) at the end-line (Fig 2).

**Table 2. Characteristics of study participants during baseline and end-line survey periods, 2022.**

| Background characteristics | Baseline | End-line |
|---|---|---|
| **Age of the mother (in years)** | | |
| 15–19 | 211 (5.0) | 183 (4.5) |
| 20–34 | 2,943 (69.0) | 2,845 (69.7) |
| 35–49 | 1,108 (26.0) | 1,053 (25.8) |
| **Maternal education** | | |
| Cannot read/no formal education | 2,365 (55.5) | 2,147 (52.6) |
| Primary school completed | 853 (20.0) | 805 (19.7) |
| Secondary and higher | 1,044 (24.5) | 1,129 (27.7)** |
| **Wealth quintile** | | |
| Most poor | 805 (18.9) | 880 (21.6) |
| More poor | 827 (19.4) | 826 (20.2) |
| Poor | 849 (19.9) | 818 (20.1) |
| Less poor | 879 (20.6) | 786 (19.3) |
| Least poor | 903 (21.2) | 771 (18.9) |
| **Number of children** | | |
| 1 | 1,043 (24.5) | 1,069 (26.2) |
| 2–3 | 1,565 (36.7) | 1,562 (38.3) |
| 4+ | 1,653 (38.8) | 1,450 (35.5)** |
| **Distance to the nearest health facility (walking distance)** | | |
| < 30 min | 1,992 (46.7) | 2,247 (55.1) |
| 30 min-1 hr | 1,500 (35.2) | 1.279 (31.3) |
| >1 hr | 771 (18.81) | 555 (13.6)* |
| **Administrative woreda** | | |
| Dembecha | 2,494 (58.5) | 2,392 (58.6) |
| Lume | 1,768 (41.5) | 1,689 (41.4) |
| **Total number of respondents** | **4,262 (100)** | **4,081 (100)** |

* Statistically significant difference (p<0.01) and

** (<0.05) between baseline and end-line surveys

**Table 3. Changes in the prevalence of neonatal illness and mother's care-seeking behaviors during baseline and end-line survey periods, June 2022.**

| Variables | Baseline | End-line |
|---|---|---|
| **Neonatal illness** | | |
| History of illness | 494 (11.6) | 345 (8.4)* |
| Formal health care seeking for ill neonates | 294 (57.6) | 313 (87.8)* |
| ***Number of sick young infants*** | ***510*** | ***357*** |
| **Neonatal severe infections (PSBI)** | | |
| Newborns with severe neonatal infection | 357 (8.4) | 250 (6.1)** |
| Neonatal infection treated with antibiotics | 210 (56.1) | 241 (91.3)* |
| ***Number of PSBI cases*** | ***374*** | ***264*** |

* Statistically significant difference (p<0.01) and

** (<0.05) between baseline and end-line surveys

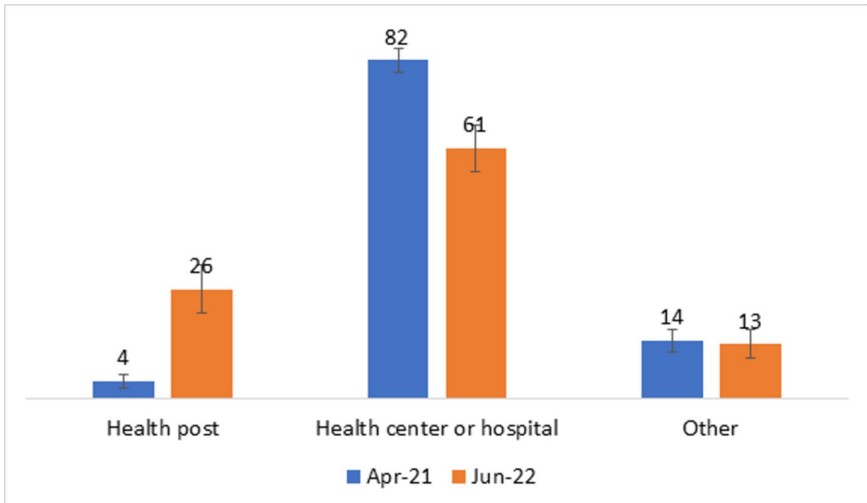

**Fig 2. Changes in the place of care-seeking during baseline and end-line survey periods, 2022.**

## Effective PSBI treatment coverage

We analyzed the cascade across the PSBI care pathway from the baseline and end-line surveys. Accordingly, the proportion of infants identified with PSBI that took any drug significantly improved from 52.9% [95% CI: 42.7–62.8] in April 2021 to 82.1% [95% CI: 75.0–87.5] in June 2022. Likewise, the proportion of severely ill young infants that took appropriate antibiotics significantly improved from 32.6% [95% CI: 25.5–40.7] in April 2021 to 61.8% [95% CI: 51.0–71.6] in June 2022 as presented in the error bar (Fig 3).

## Perceived effectiveness of the strategies

According to IDI respondents, the implementation of the adaptive strategies was helpful to 1) recover services interrupted due to COVID-19; 2) enhance the capacity of HEWs; 3) improve community awareness about the iCCM/PSBI service availability; and 4) improve the care-

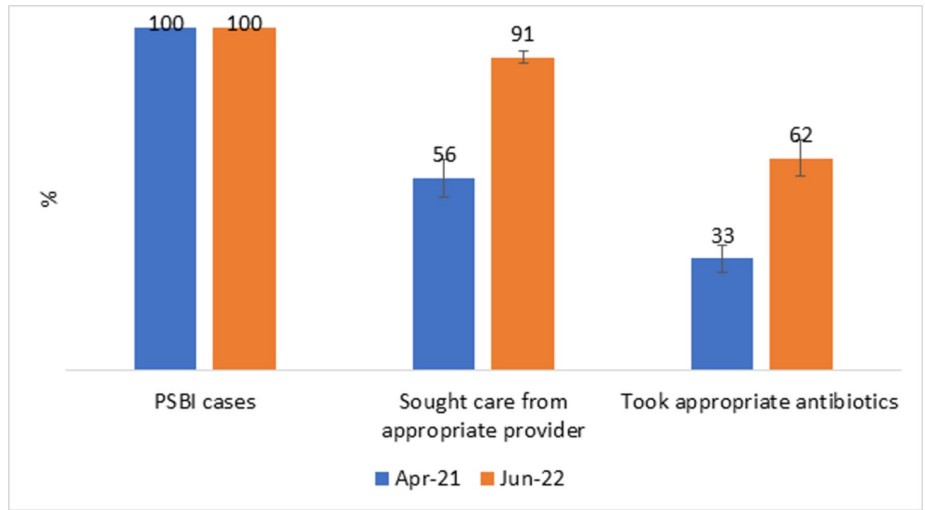

**Fig 3. Changes in PSBI treatment cascade coverage over time, 2022.**

seeking behavior of mothers for their sick young infants, which corroborates the quantitative finding.

Respondents mentioned that one of the perceived effects of this IR was a recovery of routine services from the COVID-19 pandemic.

"... the community, at some point, has stopped the service due to COVID-19. But then, we persuaded the community to resume using the service after a temporary halt." (iCCM focal person respondent)

"To our delight, now everything is turning out well or having the intended result, and most community members are using the [iCCM] services at the health posts." (iCCM Focal Person respondent)

Other effects of this investment mentioned were enhanced capacity of health centers' strengthened ability to support HEWs:

"The success achieved in boosting the capacity of HEWs is worth mentioning. Generally, the support from JSI was immense in many aspects, including the training provided for our health professionals, which was unthinkable during the COVID-19 era. More importantly, the support we received equipped us with the means to improve the capacity of HEWs." (iCCM focal person respondent)

## Ongoing implementation challenges

Despite improved health post readiness and demand generation activities to improve the uptake and delivery of iCCM/PSBI services, respondents highlighted remaining barriers impeding optimal iCCM/PSBI services, including 1) perceived low quality of service and community trust; 2) HEWs workload and shortage of HEWs; and 3) shortage of supplies.

*Low perceived quality of and trust in health post care*: HEWs who participated in the IDIs reported that communities' perceived low quality of and trust in health post care delayed care-seeking at facilities as they opted for traditional healers. They also bypassed health posts to seek care at health centers or hospitals. Respondents said:

"They [communities] usually do not believe they will get good service at health posts." (HEW respondent)

"The community members usually have limited trust in the services at health posts." (HEW respondent)

Poor health posts infrastructure also affects the healthcare-seeking behaviors of the community:

"... if you have seen the condition of the health posts you visited today, I am sure you have noticed what the roofs look like other health posts and have neither a toilet nor a fence. The rooms are full of dust. They are in bad condition, and the situations the health posts are in will make every client feel disappointed." (iCCM focal person respondent)

*Shortage of drugs*: One of the remaining challenges to delivering iCCM/PSBI services was a shortage of supplies and essential drugs. Amoxicillin dispersible tablet was a persistent challenge:

"... *the first challenge is related to the shortage of supplies and medicines. Especially the shortage of drugs has been a serious challenge for us to provide the iCCM service.*" (iCCM focal person respondent)

*Shortage of HEWs, workload, and other competing priorities*: Shortage of HEWs and workload were also reported as significant challenges for delivering iCCM/PSBI services. For

example, the health posts would be closed when HEWs conduct community and outreach-based services. One HEW respondent said:

> "For instance, in this health post, as you can see, I am alone, and when there are campaigns like for the trachoma treatment, I stay on the field for ten consecutive days. So, the health post will be closed during campaigns and other activities. Closure of health posts affects the health service utilization of community members as they become upset when they get the health post closed."

Other competing priorities of the government also affected the delivery of iCCM/PSBI services:

> "Well, the government is over-engaged with emerging tasks, and when such activities arise, we interrupt iCCM activities and move to the emergent task. So, we would rather focus on these [emergent] tasks and then have our iCCM performance evaluated." (HEW respondent)

## Discussion

The adaptive implementation strategies designed and implemented to maintain quality community-based PSBI treatment in this implementation research effectively engaged community volunteers, HEWs, and the primary health care system, and improved access to and quality delivery of PSBI treatment during the COVID-19 pandemic. The trends in the effective coverage for the appropriate management of PSBI have also significantly improved following the intervention.

The prevalence of PSBI significantly decreased over the implementation period, which is lower than in previous studies in Ethiopia and systematic review and meta-analysis studies [34, 37]. This might be attributed to the implementation strategies that promoted facility birth and essential newborn care practice, as well as community education regarding COVID-19 preventive measures. The qualitative component of the study supported this. The technical oversight and support system from woreda and PHC over community health care systems effectively engaged community volunteers and HEWs in their routine duties leading to improved linkages between communities and health facilities and delivery of PSBI treatment. In addition, facilitating the integration of PSBI management into the woreda and PHC work stream (support system, performance review, and planning and budgeting) improved fidelity and sustained implementation. Previous studies also reported that such strategies activated primary health care and community volunteers to treat PSBI [13, 26, 38].

This implementation research showed that mothers' care-seeking behavior for medical care for their sick newborns, as well as intervention-adjusted coverage of receiving appropriate antibiotics, increased significantly. While most mothers sought medical care for their severely sick newborns after the intervention, the coverage of newborns who took appropriate antibiotics is still low, as was also documented in earlier implementation studies [30]. The possible reason for the dropout across the treatment cascade could be the low competence of caregivers.

Designing strategies to improve the quality of care is a critical component of ending preventable newborn mortality and global health leaders explicitly call for the improvement in quality of healthcare services as an integral component to achieve universal health coverage of SDG targets [19]. Evidence shows low levels of treatment-seeking and community misperceptions about the severity of newborn illnesses or quality of care [14, 15], caregivers' lack of recognition of signs of newborn infection and low care-seeking prevalence [39, 40], and poor adherence to iCCM treatment guidelines [41, 42] contributed to low service quality. Strengthening the health system, improved supervision and additional training for providers, and

interventions to change the workplace culture have been tested to improve adherence to protocols and quality of health care in developing countries [43]. In addition, for higher effective coverage of PSBI treatment when referral is not feasible, social mobilization strategies targeting families and communities and client-provider interactions need to be strengthened to adhere to treatment protocols and proper counseling and adherence monitoring.

Previous implementation research in Ethiopia demonstrated a higher treatment coverage for PSBI through training health care workers, in-field technical support, performance reviews [26], community engagement [12, 26], strengthening supply chain management [26], standardization of treatment protocols at the health center and health post levels [26], policy dialogue, presence of state-level sensitization workshop, and establishment of a technical support unit [12, 26]. The qualitative component of this IR affirmed that implementation strategies, particularly the social mobilization interventions, improved communities' awareness about the availability of PSBI services in their vicinity health posts. In this IR, HEWs identified and managed about 26% of PSBI cases at the end-line, which is a significant improvement from 4% at baseline where most families sought care directly from health centers or hospitals, or sought no care at all. Access to PSBI treatment near their homes would help avert difficulties of access to care to facilities in rural areas due to terrain and long distances.

Important strategies to address access and continuum of care challenges include identification of all pregnancies and antenatal counseling visits, which promote antepartum care and birth preparedness and notification; notification of all deliveries within the first day; and early postnatal home visits to assess newborns, counsel mothers (including danger sign recognition), and refer and/or manage suspected newborn infections for identifying cases and beginning full-course of appropriate antibiotics treatment [21]. In this implementation research, HEWs made about 35 home visits per health post per month and identified an average of three infants per health post per month between April 2021 to May 2022. However, while postpartum home visitation showed improvement, it remains weak in the country. Further strengthening this link will be key for the successful implementation of PSBI case management at a community level across Ethiopia.

In Ethiopia, the iCCM/PSBI treatment program is integrated with the continuum of maternal and child health care framework from pregnancy to childhood and from household to health facilities. We have learned that integration of strategies into the woreda health system, continuous support systems, and technical oversight were critical for frontline health workers to deliver PSBI services and to create a resilient community health system to provide quality PSBI care during the pandemic. Implementation facilitation and support to the HEP could improve the delivery of PSBI management in the community when hospital referral is not feasible. As the HEP is the intermediary unit between the community and primary care facilities in a woreda health system, strong relationships with stakeholders in the community and health system are also critical for high performance. Strong technical oversight and accountability mechanisms are found to facilitate community trust, communication, and relationships, which influence HEP performance [44]. Previous studies also demonstrated that more frequent and supportive guidance and administrative support, audits with feedback, and checklist-based supervision create an enabling work environment [45, 46] and improve the performance and motivation of primary healthcare workers [47].

Despite observed improvements, remaining barriers impeded optimal iCCM/PSBI service operations, including perceived low quality of service and community trust, low frequency of home visitation, high workloads and shortage of HEWs, and lack of supplies. Most of these challenges could be resolved by woreda and primary healthcare technical support and program oversight. Strong technical oversight and accountability mechanisms were found to foster community trust, effective communication, and positive relationships; all of which contribute

to improved health system performance [44]. To ensure successful implementation and optimal integration of management of PSBI within the health system, several critical factors need to be addressed. These include providing continued coaching and support systems for frontline health workers, ensuring technical oversight from the district/county health teams, and maintaining an adequate supply of essential commodities. Accordingly, the introduction of continuous audit and feedback systems on the quality of case management, continuous supply of essential drugs, and strategies to alleviate HEWs' workload to deliver quality services should be prioritized.

Measuring trends in coverage for the appropriate management of PSBI by place of treatment (health facility or community) and type of provider are well recognized to assess the quality of the iCCM program to address critical bottlenecks, which limit capacity to provide effective care, including access to and use of care. Although effective coverage has been increasingly used in the evaluation of maternal and child health programs, population-level data on the place of treatment and the type of health provider are scarce in LMICs. To our knowledge, this is the first study that estimates both crude and effective coverage of PSBI management. Another strength of the study is the participatory design and implementation with active engagement of stakeholders that foster ownership and integration of strategies. However, we could not assess the full spectrum of treatment cascade across the treatment continuum, nor estimate input-adjusted or quality-adjusted coverage. As such, linking data from households and health facilities could produce meaningful estimates of coverage that could help to access the input and quality-adjusted coverages [35]. The study, based on health post records, revealed a substantial increase in the monthly treatment rate by HEWs due to project support. However, limitations such as the absence of a comparison group and pre-COVID-19 data make it challenging to attribute outcomes solely to the intervention. Time-invariant confounders such as changes in the background characteristics including maternal education, parity, and distance to facility or other developmental inputs (road, health facility construction, etc.) that influence maternal and newborn health could bias the observed outcome. However, as this was an implementation research, we were continuously monitoring and adapting, and/or modifying strategies to respond to contextual factors. As such, the observed effect could be explained by the implementation intensity of the strategies rather than the changes in developmental inputs.

## Conclusions

The COVID-19 adaptive implementation strategies included training and coaching, creation of support systems and technical support units, supply strengthening of essential commodities, and community awareness creation about PSBI and COVID-19; these strategies led to improved care-seeking behavior of mothers for their sick young infants and improved the quality of PSBI management. Technical oversight and support systems as well as integration of PSBI treatment into the woreda and PHC system were critical for frontline health workers to deliver PSBI services and create a resilient community health system to provide quality PSBI care during the pandemic. Continuous audit and feedback systems on the quality of case management, continuous supply of essential drugs, strong technical oversight and accountability, and strategies to alleviate HEWs' workload to deliver quality services should be prioritized. In addition, social mobilization strategies targeting families and communities, as well as client-provider interactions need to be strengthened to further improve effective coverage of community-based management of PSBI when referral is not feasible.

## Supporting information

**S1 File. Survey questionnaire.** A household survey questionnaire we used to collect information from study participants. The first sheet contains variable definitions (data dictionary) in English and local languages, and the second sheet contains variable answer choices.
(XLSX)

**S2 File. In-depth interview guides.** These are interview guides for service providers.
(DOCX)

**S3 File. Survey dataset.** This is household survey data with variables and the values we used for this analysis.
(CSV)

## Acknowledgments

The implementation of this survey would not have been possible without the support of Ethiopia's Ministry of Health, Amhara and Oromia Regional Health Bureaus, National Child Health Technical Working Group, National reproductive, maternal, newborn, child, adolescent health and nutrition Research Advisory Council Child Health and Immunization group, and other development partners (Transform Primary Health Care, Fenot project, UNICEF, KOICA, and USAID). We acknowledge the interviewers and the supervisors for their hard work, dedication, and for accomplishing the fieldwork on schedule. Finally, we take this opportunity to extend our gratitude to all study participants who took their time to respond to the survey questionnaires and provide us with invaluable information. Alexandra Kamberos is well-acknowledged for editing the manuscript.

## Author Contributions

**Conceptualization:** Gizachew Tadele Tiruneh, Nebreed Fesseha, Dessalew Emaway, Wuleta Betemariam, Tsinuel Girma Nigatu, Lisa Ruth Hirschhorn.

**Data curation:** Gizachew Tadele Tiruneh, Nebreed Fesseha, Dessalew Emaway, Wuleta Betemariam, Hema Magge, Lisa Ruth Hirschhorn.

**Formal analysis:** Gizachew Tadele Tiruneh, Tsinuel Girma Nigatu, Lisa Ruth Hirschhorn.

**Funding acquisition:** Gizachew Tadele Tiruneh, Nebreed Fesseha, Dessalew Emaway, Wuleta Betemariam, Hema Magge, Lisa Ruth Hirschhorn.

**Investigation:** Gizachew Tadele Tiruneh, Nebreed Fesseha, Tsinuel Girma Nigatu, Lisa Ruth Hirschhorn.

**Methodology:** Gizachew Tadele Tiruneh, Nebreed Fesseha, Tsinuel Girma Nigatu, Hema Magge, Lisa Ruth Hirschhorn.

**Project administration:** Gizachew Tadele Tiruneh, Nebreed Fesseha, Dessalew Emaway, Wuleta Betemariam, Hema Magge, Lisa Ruth Hirschhorn.

**Supervision:** Lisa Ruth Hirschhorn.

**Visualization:** Gizachew Tadele Tiruneh, Nebreed Fesseha, Dessalew Emaway, Wuleta Betemariam, Hema Magge, Lisa Ruth Hirschhorn.

**Writing – original draft:** Gizachew Tadele Tiruneh, Dessalew Emaway, Wuleta Betemariam, Tsinuel Girma Nigatu, Hema Magge, Lisa Ruth Hirschhorn.

**Writing – review & editing:** Gizachew Tadele Tiruneh, Nebreed Fesseha, Dessalew Emaway, Wuleta Betemariam, Tsinuel Girma Nigatu, Hema Magge, Lisa Ruth Hirschhorn.

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
