## [Decision Letter · Decision Letter 0]

3 Oct 2023

PONE-D-23-12889Effect of community-based newborn care implementation strategies on access to and effective coverage of possible serious bacterial infection (PSBI) treatment for sick young infants during COVID-19 pandemicPLOS ONE

Dear Dr. Tiruneh,

Thank you for submitting your manuscript to PLOS ONE. After careful consideration, we feel that it has merit but does not fully meet PLOS ONE’s publication criteria as it currently stands. Therefore, we invite you to submit a revised version of the manuscript that addresses the points raised during the review process.

We look forward to receiving your revised manuscript.

Kind regards,

Hassen Mosa, Msc

Academic Editor

PLOS ONE

Journal Requirements:

https://bmchealthservres.biomedcentral.com/articles/10.1186/s12913-022-08945-9

https://publications.jsi.com/JSIInternet/Inc/Common/_download_pub.cfm?id=24907&lid=3

In your revision ensure you cite all your sources (including your own works), and quote or rephrase any duplicated text outside the methods section. Further consideration is dependent on these concerns being addressed.

"We thank the Bill & Melinda Gates Foundation for funding this implementation research."

"The article write-up and publication fee were supported by the Bill & Melinda Gates Foundation, Investment Number INV-024241. JSI Research & Training Institute, Inc. has helped us in the form of salaries for authors [GT, NF, DE, and WB]. However, none of the funders played a role in the study design, data collection, analysis, decision to publish, or manuscript preparation."

"The authors declare that they have no competing interests. The authors [GT, NF, DE, and WB] work for JSI Research & Training Institute, Inc., a commercial company. We declare that this commercial affiliation does not alter our adherence to the journal's policies on sharing data and materials. One of the authors of this manuscript (HM) works for the Gates Foundation. We would like to declare that we do not have any conflict of interest with the Gates Foundation-paid staff in preparing this manuscript."

5. Please upload a new copy of Figure 1 as the detail is not clear. Please follow the link for more information: " ext-link-type="uri" xlink:type="simple">https://blogs.plos.org/plos/2019/06/looking-good-tips-for-creating-your-plos-figures-graphics/"
https://blogs.plos.org/plos/2019/06/looking-good-tips-for-creating-your-plos-figures-graphics

Reviewers' comments:

Reviewer's Responses to Questions

**Comments to the Author**

1. Is the manuscript technically sound, and do the data support the conclusions?

Reviewer #1: Yes

Reviewer #2: Yes

2. Has the statistical analysis been performed appropriately and rigorously? 

Reviewer #1: Yes

Reviewer #2: Yes

3. Have the authors made all data underlying the findings in their manuscript fully available?

Reviewer #1: Yes

Reviewer #2: Yes

4. Is the manuscript presented in an intelligible fashion and written in standard English?

Reviewer #1: Yes

Reviewer #2: Yes

5. Review Comments to the Author

Reviewer #1: Methods section:

1. Around line 139, it would have been good if they clearly describe the roles and responsibilities of each in detail so that readers understand the different categories of role players (like the HEWs did this and that, implementers did this and that, etc.). Plus, the intervention package should have also been elaborated so that one understands what has been done differently that led to the changes in the outcome variables.

2. Line 147: Why did you use prevalence estimate of 5%? There are different figures as to the prevalence (estimation) of PSBI in the first 2months of life.

3. Outcome measures: line 182: receipt of appropriate antibiotics: What does this mean? Just a dose, 2 doses, 5 doses or what? Plus, would you call appropriate/effective if an infant receives amoxicillin or gentamicin alone? What about those who received other antibiotics like ampicillin?

Results and discussion sections:

4. Line 220: Could the difference in the background characteristics have an effect on the outcome variables? Can you highlight this in the discussion as well?

5. Line 226-232: Instead of the change in the prevalence of PSBI, did you try look at the rate of detection (compared to the estimated/expected) of PSBI before and after the implementation research which could also indicate the changes with regard to the IR?

6. Changes in the prevalence of PSBI and care seeking behaviour: What specific strategies did you carry out which have led to these change? Promoting institutional deliveries and improving ENCs have already been implemented for long time but with no significant changes and I think it is well known that the intensive COVID 19 prevention strategies did last only for few weeks/months specially in the rural areas. So, attributing these to the factors that led to changes is somehow implausible.

7. Changes in the place of care seeking: Is this really because the care givers wanted to get the service at the HPs or because they are forced to get the service there? Did you try to understand the reason for this from the care givers as well as the HEWs/HCWs perspectives? I think the recommendation for PSBI treatment is still hospitalization and treatment at HP/HC is only if referral is not possible/feasible.

Conclusion/recommendations:

8. These challenges you have figured out have long been identified by the several prior studies as well. What should be done differently to tackle these challenges for the subsequent successful implementation of such strategies?

Reviewer #2: The paper is well written and can be published after some minor comments have been addressed.

1. Clear Explanation of iCCM strategy would be useful specifically with the roles of different facilities and providers - especially the role of HEW vs WDA and HEW vs. Nurse. How is this strategy different from the current implementation strategies applicable in the country.

2. While barriers and strategies have been referred to in the other paper by the authors, a better explanation of these in figure 1, would be useful to understand what strategies were implemented and what worked and others did not to increase the effective coverage.

3. WHile extensive household survey seems to have been done, not much results of the survey can be seen in the paper.

Some other comments have been added in text boxes in the attached paper.

6. PLOS authors have the option to publish the peer review history of their article (what does this mean?). If published, this will include your full peer review and any attached files.

Reviewer #1: No

Reviewer #2: **Yes: **Charu C Garg

---

## [Author Response · Author response to Decision Letter 0]

19 Oct 2023

Point-by-point response

Cover letter

PONE-D-23-12889

Effect of community-based newborn care implementation strategies on access to and effective coverage of possible serious bacterial infection (PSBI) treatment for sick young infants during COVID-19 pandemic

Dear editors,

We, the authors, would like to thank the reviewers and the editor for their valuable comments. Our point-by-point responses to the reviewers and editor are below each of the comments in italics. We also reviewed to ensure that this version of the manuscript conforms to the journal style. 

Journal Requirements:

Response: Thank you for the comments. The manuscript is formatted according to the journal style requirements. 

https://bmchealthservres.biomedcentral.com/articles/10.1186/s12913-022-08945-9

https://publications.jsi.com/JSIInternet/Inc/Common/_download_pub.cfm?id=24907lid=3

In your revision ensure you cite all your sources (including your own works), and quote or rephrase any duplicated text outside the methods section. Further consideration is dependent on these concerns being addressed.

Response: Our apologies and thank you for alerting us. In this version, we revised it accordingly and have checked to ensure that any repeated language is marked as quote and referenced. 

"We thank the Bill Melinda Gates Foundation for funding this implementation research."

"The article write-up and publication fee were supported by the Bill Melinda Gates Foundation, Investment Number INV-024241. JSI Research Training Institute, Inc. has helped us in the form of salaries for authors [GT, NF, DE, and WB]. However, none of the funders played a role in the study design, data collection, analysis, decision to publish, or manuscript preparation."

Response: Thank you for the comments. The funding information is removed from the manuscript and the Acknowledgment section. The Funding Statement submitted in the online submission form is correct. 

"The authors declare that they have no competing interests. The authors [GT, NF, DE, and WB] work for JSI Research Training Institute, Inc., a commercial company. We declare that this commercial affiliation does not alter our adherence to the journal's policies on sharing data and materials. One of the authors of this manuscript (HM) works for the Gates Foundation. We would like to declare that we do not have any conflict of interest with the Gates Foundation-paid staff in preparing this manuscript."

Response: Thank you for the comments. It is now updated as follows;

"The authors declare that they have no competing interests. The authors [GT, NF, DE, and WB] work for JSI Research Training Institute, Inc., a commercial company. This does not alter our adherence to PLOS ONE policies on sharing data and materials. One of the authors of this manuscript (HM) works for the Gates Foundation. We would like to declare that we do not have any conflict of interest with the Gates Foundation-paid staff in preparing this manuscript."

5. Please upload a new copy of Figure 1 as the detail is not clear. Please follow the link for more information: https://blogs.plos.org/plos/2019/06/looking-good-tips-for-creating-your-plos-figures-graphics/" https://blogs.plos.org/plos/2019/06/looking-good-tips-for-creating-your-plos-figures-graphics

Response: Thank you for the comments. The revised version is uploaded. 

Response: Thank you for the comments. The references are reviewed. 

Reviewers' comments:

Review Comments to the Author

Reviewer #1: Methods section:

1. Around line 139, it would have been good if they clearly describe the roles and responsibilities of each in detail so that readers understand the different categories of role players (like the HEWs did this and that, implementers did this and that, etc.). Plus, the intervention package should have also been elaborated so that one understands what has been done differently that led to the changes in the outcome variables.

Response: Thank you for the valid comments. We presented the strategies, details of the activities, and roles of stakeholders in tabular form (see additional Table 1, page 8)

2. Line 147: Why did you use prevalence estimate of 5%? There are different figures as to the prevalence (estimation) of PSBI in the first 2months of life.

Response: We agree that different studies have reported different values of prevalence of PSBI that range from 5-6% at baseline and 11-13% at end line (Berhanu et al., 2020). A systematic review and meta-analysis that included data from 22 studies reported PSBI incidence risk estimate of 6% in the sub-Saharan Africa region and overall pooled estimate of 7.6% (Seale et al., 2014). Some studies reported more than 10% prevalence which could be due to the inclusion of local infection signs (Puri et al., 2021). We took 5% prevalence based on the baseline estimate of a large household study in Ethiopia (Berhanu et al., 2020) as our primary objective was to see the effect of the IR on improved care-seeking for PSBI treatment Our sample size was determined based on the prevalence of care-seeking for PSBI, 56% not 5% of PSBI. We used the 5% prevalence estimate to check whether we could get an adequate sample of mothers seeking care for PSBI in the study areas. 

3. Outcome measures: line 182: receipt of appropriate antibiotics: What does this mean? Just a dose, 2 doses, 5 doses or what? Plus, would you call appropriate/effective if an infant receives amoxicillin or gentamicin alone? What about those who received other antibiotics like ampicillin?

Response: In this study, appropriate treatment for PSBI is defined as the recipient of antibiotics for the treatment of PSBI according to the national iCCM/IMNCI standards of treatment which includes gentamicin injection and amoxicillin dispersible tablet. Accordingly, the intervention-adjusted coverage of PSBI treatment was therefore calculated counting appropriate treatment as recipient of these antibiotics irrespective of the completion of the full course treatment. We also did not count those who took inappropriate treatment or unnecessary antibiotics such as tetracycline eye ointments, antimalarial drugs, ORS, vitamin A supplements, and/ or traditional remedies. While adherence-adjusted would include completion of treatment that is gentamicin injection for 2 days and amoxicillin tablet for 7 days for full course completion of the treatment which is not included in this analysis. The manuscript is revised accordingly to clarify this. See page 11, lines 176-189.

Results and discussion sections:

4. Line 220: Could the difference in the background characteristics have an effect on the outcome variables? Can you highlight this in the discussion as well?

Response: Definitely, differences in the background characteristics, or other developmental inputs like road construction that influence MNH could be time-invariant confounders and bias the observed outcome. These are now discussed in the discussion section of the revised version (Page 23, lines 403-409) of the manuscript as presented below.

“Time-invariant confounders such as changes in the background characteristics including maternal education, parity, and distance to facility over time or other developmental inputs (road, health facility construction, etc.) that influence maternal and newborn health could bias the observed outcome. However, as this was an implementation research, we were continuously monitoring and adapting and/or modifying strategies to respond to contextual factors. As such, the observed effect could be explained by the implementation intensity of the strategies than the changes in developmental inputs.” 

5. Line 226-232: Instead of the change in the prevalence of PSBI, did you try look at the rate of detection (compared to the estimated/expected) of PSBI before and after the implementation research which could also indicate the changes with regard to the IR?

Response: Thank you again for the valid comments. We tracked monthly detection rates of PSBI from facility records and monitor the effect of IR on PSBI treatment. In this paper, we presented the changes in the prevalence and care-seeking behavior of mothers for PSBI (from 2 rounds of household surveys). 

6. Changes in the prevalence of PSBI and care seeking behaviour: What specific strategies did you carry out which have led to these change? Promoting institutional deliveries and improving ENCs have already been implemented for long time but with no significant changes and I think it is well known that the intensive COVID 19 prevention strategies did last only for few weeks/months specially in the rural areas. So, attributing these to the factors that led to changes is somehow implausible.

Response: The support system, technical oversight from the HC and woreda health office were designed to help HEWs and community volunteers to engage into their routine duties. In addition, this approach of mapping strategies to address identified challenges facilitated integration of PSBI treatment into the woreda and PHC work stream (support system, performance review, and planning and budgeting). We have included the following sentences on the manuscript to make it clear. 

“The qualitative component of the study supported this. The technical oversight and support system from woreda and PHC over community health care systems effectively engaged community volunteers and HEWs into their routine duties leading to improved linkages between communities and health facilities and delivery of PSBI treatment. In addition, facilitating integration of PSBI management into the woreda and PHC work stream (support system, performance review, and planning and budgeting) improved fidelity and sustained implementation. Previous studies also reported that such strategies activated primary health care and community volunteers to treat PSBI (Avan et al., 2021; Berhane et al., 2021; Mukhopadhyay et al., 2021).” Page 19, lines 313-320

7. Changes in the place of care seeking: Is this really because the care givers wanted to get the service at the HPs or because they are forced to get the service there? Did you try to understand the reason for this from the care givers as well as the HEWs/HCWs perspectives? I think the recommendation for PSBI treatment is still hospitalization and treatment at HP/HC is only if referral is not possible/feasible.

Response: Both the qualitative and quantitative interviews with mothers revealed it is the communities’ awareness about the availability of PSBI services in their vicinity health posts that improved care seeking at HPs. 

Conclusion/recommendations:

8. These challenges you have figured out have long been identified by the several prior studies as well. What should be done differently to tackle these challenges for the subsequent successful implementation of such strategies?

Response: Thank you again for the valid comments. Optimizing the integration of PSBI management into the woreda and PHC health system and continuous technical oversight would be helpful. Strong technical oversight and accountability mechanisms were found to foster community trust, effective communication, and positive relationships; all of which contribute to improved health system performance (Kok et al., 2015). To ensure successful implementation and optimal integration of management of PSBI within the health system, several critical factors need to be addressed. These include providing continued coaching and support systems for frontline health workers, ensuring technical oversight from the district/county health teams, and maintaining an adequate supply of essential commodities. The paragraph is edited accordingly, page 22, lines 378-390.

“Despite observed improvements, remaining barriers impeded optimal iCCM/PSBI service operations, including perceived low quality of service and community trust, low frequency of home visitation, high workloads and shortage of HEWs, and lack of supplies. Most of these challenges could be resolved by woreda and primary healthcare technical support and program oversight. Strong technical oversight and accountability mechanisms were found to foster community trust, effective communication, and positive relationships; all of which contribute to improved health system performance (Kok et al., 2015). To ensure successful implementation and optimal integration of management of PSBI within the health system, several critical factors need to be addressed. These include providing continued coaching and support systems for frontline health workers, ensuring technical oversight from the district/county health teams, and maintaining an adequate supply of essential commodities. Accordingly, the introduction of continuous audit and feedback systems on the quality of case management, continuous supply of essential drugs, and strategies to alleviate HEWs’ workload to deliver quality services should be prioritized.” 

Reviewer #2: The paper is well written and can be published after some minor comments have been addressed.

1. Clear Explanation of iCCM strategy would be useful specifically with the roles of different facilities and providers - especially the role of HEW vs WDA and HEW vs. Nurse. How is this strategy different from the current implementation strategies applicable in the country.

Response: Thanks a lot for the comments. The following paragraph is added to explain the iCCM strategy and the roles of HEWs and CHWs.

 “HEWs and WDAs are responsible for educating mothers about early recognition of illness in newborns, and prompt care-seeking assessing danger signs in newborns during postnatal home visits (MOH, 2017), In addition, community-based treatment of PSBI provided by HEWs, integrated with the iCCM program. HEWs are trained to identify newborn infections and provide treatment when referral is not feasible.” Page 6, lines 108-112

2. While barriers and strategies have been referred to in the other paper by the authors, a better explanation of these in figure 1, would be useful to understand what strategies were implemented and what worked and others did not to increase the effective coverage.

Response: Thank you for the valid comments. We have presented the strategies, details of the activities, and roles of role players in tabular form (see additional Table 1)

3. WHile extensive household survey seems to have been done, not much results of the survey can be seen in the paper.

Response: Thank you again for the comments. The study interviewed women about their sick newborn and about their own health during pregnancy, childbirth, and postpartum period. As the primary objective of the study was averting the impact of COVID-19 and other system bottlenecks on newborn health, in this paper, we presented the strategies designed and implemented, effect of the strategies of PSBI treatment, ongoing challenges for the treatment of PSBI, and changes observed in the uptake of PSBI treatment. 

Some other comments have been added in text boxes in the attached paper.

• Line 191- would be useful to define appropriate treatment. Is it 7 days of antibiotics or partial treatment os also considered appropriate?

Response: In this study, appropriate treatment for PSBI is defined as the recipient of antibiotics for the treatment of PSBI according to the national iCCM/IMNCI standards of treatment which includes gentamicin injection and amoxicillin dispersible tablet. Accordingly, the intervention-adjusted coverage of PSBI treatment was therefore calculated counting appropriate treatment as recipient of these antibiotics irrespective of the completion of the full course treatment. We also did not count those who took inappropriate treatment or unnecessary antibiotics such as tetracycline eye ointments, antimalarial drugs, ORS, vitamin A supplements, and/ or traditional remedies. While adherence-adjusted would include completion of treatment that is gentamicin injection for 2 days and amoxicillin tablet for 7 days for full course completion of the treatment which is not included in this analysis. The manuscript is revised accordingly to clarify this. See page 11, lines 176-189.

• It would be useful to show in a table format challenges identified in baseline, implementation strategies / interventions adopted, which strategies worked the most to improve effective coverage. at individual, HHs and community level

Response: Thank you for the valid comments. We presented the strategies, details of the activities, and roles of stakeholders in tabular form (see Table 1, page 8)

• Discussion: Clearly state the strategies and interventions that helped to remove key barriers and why some barriers continued to pursue such that coverage of newborns who took appropriate antibiotics was still low.

Response: Thank you for the valid comments. We presented the strategies, details of the activities, and roles of stakeholders in tabular form (see Table 1, page 8) page 20, lines 323-334. In addition, the continued challenges for the treatment of PSBI are discussed. Page 22, lines 392-404

• Reference: Please also add Mukhopadhyay R, Arora NK, Sharma PK, Dalpath S, Limbu P, Kataria G, et al. (2021) Lessons from implementation research on community management of Possible Serious Bacterial Infection (PSBI) in young infants (0-59 days), when the referral is not feasible in Palwal district of Haryana, India. PLoS ONE 16(7): e0252700. https://doi.org/10.1371/journal. pone.0252700

Response: Thank you for sharing the article. We consulted it and used the key findings in our discussion. (Avan et al., 2021; Berhane et al., 2021; Mukhopadhyay et al., 2021).” Page 19, lines 313-320

---

## [Decision Letter · Decision Letter 1]

27 Feb 2024

PONE-D-23-12889R1Effect of community-based newborn care implementation strategies on access to and effective coverage of possible serious bacterial infection (PSBI) treatment for sick young infants during COVID-19 pandemicPLOS ONE

Dear Dr. Tiruneh,

Thank you for submitting your manuscript to PLOS ONE. After careful consideration, we feel that it has merit but does not fully meet PLOS ONE’s publication criteria as it currently stands. Therefore, we invite you to submit a revised version of the manuscript that addresses the points raised during the review process.

We look forward to receiving your revised manuscript.

Kind regards,

Khin Thet Wai, MBBS, MPH, MA

Academic Editor

PLOS ONE

Journal Requirements:

Reviewers' comments:

Reviewer's Responses to Questions

**Comments to the Author**

1. If the authors have adequately addressed your comments raised in a previous round of review and you feel that this manuscript is now acceptable for publication, you may indicate that here to bypass the “Comments to the Author” section, enter your conflict of interest statement in the “Confidential to Editor” section, and submit your "Accept" recommendation.

Reviewer #2: All comments have been addressed

Reviewer #3: (No Response)

2. Is the manuscript technically sound, and do the data support the conclusions?

Reviewer #2: Partly

Reviewer #3: Partly

3. Has the statistical analysis been performed appropriately and rigorously? 

Reviewer #2: I Don't Know

Reviewer #3: No

4. Have the authors made all data underlying the findings in their manuscript fully available?

Reviewer #2: Yes

Reviewer #3: (No Response)

5. Is the manuscript presented in an intelligible fashion and written in standard English?

Reviewer #2: Yes

Reviewer #3: Yes

6. Review Comments to the Author

Reviewer #2: (No Response)

Reviewer #3: Thank you for your invitation

I have some reservation on the manuscripts:

Authors are trying to justify that the emergency adaptive PSBI treatment implementation strategies have changed a lot regarding treatment seeking and even time of seeking. Nonetheless, the authors did not statistically or plausibly justify the changes for the following reasons:

1. It was obvious that the overall use of essential service has declined in those year 2020 and 21, and expected to rise toward the end of 2022 as the health system started no reconsider the correction of the disruption of service at that time. What is your plausible (statistical and qualitative) evidence on this?

2. There may be a lot of factors leading to the reported changes- the authors did not present adequate evidence to plausibly confirm that the changes due to their adaptive implementation strategies. Time, duration, logic, ratio, dose, and plausibility of the change not well convincing. Time- what if the difference in time of data collection (April) and June months is a reason for increase of decline of the PSBI? Other programs could also have contributed for the changes.

3. The HEWs are practically overburdened with multiple tasks, and how do you justify such huge change?

4. The authors did not present other relevant domains of RE-AIM model to support the reported changes. example, what was the coverage of their intervention? and the adaption (individual and health system) dimensions?

Other comments

Is that true that the WDA in Ethiopia are responsible to educate neonatal danger signs- and what evidence can be presented to assure this statement in the background?

7. PLOS authors have the option to publish the peer review history of their article (what does this mean?). If published, this will include your full peer review and any attached files.

Reviewer #2: No

Reviewer #3: No

---

## [Author Response · Author response to Decision Letter 1]

5 Mar 2024

A point-by-point response to reviewers

PONE-D-23-12889R2

Effect of community-based newborn care implementation strategies on access to and effective coverage of possible serious bacterial infection (PSBI) treatment for sick young infants during COVID-19 pandemic

Dear Editor,

We, the authors, would like to thank the reviewers for their valuable comments. Our point-by-point responses to the reviewers and editor are below each comment. We also reviewed to ensure that this manuscript version conforms to the journal style. 

Reviewer Comments:

I have some reservation on the manuscripts:

Authors are trying to justify that the emergency adaptive PSBI treatment implementation strategies have changed a lot regarding treatment seeking and even time of seeking. Nonetheless, the authors did not statistically or plausibly justify the changes for the following reasons:

1. It was obvious that the overall use of essential service has declined in those year 2020 and 21, and expected to rise toward the end of 2022 as the health system started no reconsider the correction of the disruption of service at that time. What is your plausible (statistical and qualitative) evidence on this?

Response: Thank you for the valid comments. Sure, in Ethiopia, the uptake of essential health services has been disrupted by the pandemic. But essential service delivery has been restored within a few months after the pandemic, particularly, in rural communities. The project abstracted longitudinal time series data from health post records and explored the trends in the monthly rate of PSBI treatment by HEWs pre-intervention, immediate, and after-project support effects. This interrupted time-series analysis showed the monthly rate increased by 4.1%, with an annual increase of 4.6%, higher than expected without the project's PSBI treatment strategies. This has been published elsewhere (Gizachew Tadele Tiruneh et al., 2024). 

Still, we acknowledged that one of the limitations of this study is the lack of comparison group and pre-COVID-19 data to examine causality. As such, in this version, we included the following sentences in the limitations of the study. “The study, based on health post records, revealed a substantial increase in the monthly treatment rate by HEWs due to project support. However, limitations such as the absence of a comparison group and pre-COVID-19 data make it challenging to attribute outcomes solely to the intervention.” See on page 24, lines 418-421.

2. There may be a lot of factors leading to the reported changes- the authors did not present adequate evidence to plausibly confirm that the changes due to their adaptive implementation strategies. Time, duration, logic, ratio, dose, and plausibility of the change not well convincing. Time- what if the difference in time of data collection (April) and June months is a reason for increase of decline of the PSBI? Other programs could also have contributed for the changes.

Response: Thank you again for the comments. As described above, the time series data from health post records, controlling seasonal variations, revealed a substantial increase in the monthly treatment rate by HEWs due to project support. However , the pre-post only design nature of the study could not able to prove attribution of the strategies. In addition, as you righly said, time-varying unobserved confounders would bias the observed treatment effects. In this regard, however, as this an IR, we were contiously tracking other programs such as presence of NGO support for maternal and child healthas a facilitator in one woreda and conxtual factors such as civil conflict in another woreda were captured and made strategy adaptiaions to address these challenges to maitain high fidelity implementation of the straegies, published elsewhere (Tiruneh, Nigatu, Magge, Hirschhorn, 2022). For this purpose, we used the IRLM framework to capture these emerging or persistent challenges and adapted strategies to articulate the mechanisms of action and link them to outcomes (Tiruneh, Nigatu, Magge, Hirschhorn, 2022).

3. The HEWs are practically overburdened with multiple tasks, and how do you justify such huge change?

Response: Sure, HEWs and other health workers in the primary care system are often reported being overburdened with multiple tasks. However, time motion and efficiency studies demonstrated that substantial time is spent waiting for clients (Arsenault et al., 2021; Tilahun et al., 2017). 

We learned that performance of the HEW could be significantly improved if they are supported, mentored and supplied with necessary supplies as well as expanding service delivery modalities and demand creation activities. As such, the project implemented on-site coaching and training for HEWs, strengthened the supply chain and support system, and integrated the management of possible severe bacterial infection (PSBI) into the woreda health system work stream. This approach actively engaged HEWs, their supervisors, and WDAs in the delivery of PSBI treatment.

4. The authors did not present other relevant domains of RE-AIM model to support the reported changes. example, what was the coverage of their intervention? and the adaption (individual and health system) dimensions?

Response: Thank you for the valid comments. The following paragraphs are included on line 145-159, pages 9-10. “The strategies were implemented effectively, ensuring the continued delivery of PSBI services while improving mothers' care-seeking behavior for their sick young infants. Technical support units comprised of iCCM trained personnel from under-five clinics of health centers, health center directors, and woreda health office child health officers were established. They provided on-site training and coaching to HEWs through supportive supervision, facilitated performance reviews, coordinated referrals, provided feedback to HEWs, identified implementation challenges, developed interventions in collaboration with stakeholders, and supported community sensitization efforts [16]. Monitoring data revealed that 84% of HEWs treated at least one sick young infant in the previous six months. Between April 2021 and May 2022, HEWs conducted 32,428 home visits, identifying 3,099 newborns, approximately three infants per health post per month. During the same period, HEWs assessed 1,335 severe young infant illness cases, representing an 85% coverage rate. 

Implementation strategies were largely integrated into community-level healthcare worker workstreams, woreda reports, and the MOH’s annual planning and forecasting of iCCM/ PSBI commodities for 2023 and 2024.”

Other comments

Is that true that the WDA in Ethiopia are responsible for educating neonatal danger signs- and what evidence can be presented to assure this statement in the background?

Response: Sure, according to the national guidance, WDAs with support from HEWs are expected to conduct home visits and educate mothers about maternal and newborn danger signs, identify sick newborns, and link to the facility (MOH, 2012). Accordingly, in settings where WDAs are well supported by the HEWs, they identified and referred mothers and newborns and linked them to HEP and/or health facilities. This resulted in improved uptake of institutional deliveries and newborn care practices such as clean cord care for newborns, thermal care for newborns, and immediate initiation of breastfeeding (Karim et al., 2018; Zufan Abera Damtew et al., 2018). However, in most cases, postpartum and newborn health care is less prioritized in the country (Tiruneh, Worku, Berhane, Betemariam, Demissie, 2020) and WDAs have less engagement in newborn and child health (Ashebir, Medhanyie, Mulugeta, Persson, Berhanu, 2020).

---

## [Decision Letter · Decision Letter 2]

7 Mar 2024

Effect of community-based newborn care implementation strategies on access to and effective coverage of possible serious bacterial infection (PSBI) treatment for sick young infants during COVID-19 pandemic

PONE-D-23-12889R2

Dear Dr. Tiruneh,

We’re pleased to inform you that your manuscript has been judged scientifically suitable for publication and will be formally accepted for publication once it meets all outstanding technical requirements.

Kind regards,

Khin Thet Wai, MBBS, MPH, MA

Academic Editor

PLOS ONE

Additional Editor Comments (optional):

All comments are adequately addressed.

Reviewers' comments:

Reviewer's Responses to Questions

**Comments to the Author**

1. If the authors have adequately addressed your comments raised in a previous round of review and you feel that this manuscript is now acceptable for publication, you may indicate that here to bypass the “Comments to the Author” section, enter your conflict of interest statement in the “Confidential to Editor” section, and submit your "Accept" recommendation.

Reviewer #3: (No Response)

2. Is the manuscript technically sound, and do the data support the conclusions? 

Reviewer #3: (No Response)

3. Has the statistical analysis been performed appropriately and rigorously? 

Reviewer #3: (No Response)

4. Have the authors made all data underlying the findings in their manuscript fully available?

Reviewer #3: (No Response)

5. Is the manuscript presented in an intelligible fashion and written in standard English?

Reviewer #3: (No Response)

6. Review Comments to the Author

Reviewer #3: (No Response)

7. PLOS authors have the option to publish the peer review history of their article (what does this mean?). If published, this will include your full peer review and any attached files.

Reviewer #3: No

---

## [Editor Report · Acceptance letter]

13 Mar 2024

PONE-D-23-12889R2 

PLOS ONE

Dear Dr. Tiruneh, 

I'm pleased to inform you that your manuscript has been deemed suitable for publication in PLOS ONE. Congratulations! Your manuscript is now being handed over to our production team.

Kind regards, 

on behalf of

Dr. Khin Thet Wai 

Academic Editor

PLOS ONE